# A Balanced Dietary Ratio of n-6:n-3 Polyunsaturated Fatty Acids Exerts an Effect on Total Fatty Acid Profile in RBCs and Inflammatory Markers in Subjects with Obesity

**DOI:** 10.3390/healthcare11162333

**Published:** 2023-08-18

**Authors:** Karina Gonzalez-Becerra, Elisa Barron-Cabrera, Jose F. Muñoz-Valle, Nathaly Torres-Castillo, Juan J. Rivera-Valdes, Roberto Rodriguez-Echevarria, Erika Martinez-Lopez

**Affiliations:** 1Instituto de Investigación en Genética Molecular, Centro Universitario de la Ciénega, Universidad de Guadalajara, Av. Universidad 1115, Ocotlán 47810, Jalisco, Mexico; karina.gbecerra@academicos.udg.mx; 2Facultad de Ciencias de la Nutrición y Gastronomía, Universidad Autónoma Sinaloa, Av Cedros y Calle Sauces S/N, Culiacán 80010, Sinaloa, Mexico; elisabarron@uas.edu.mx; 3Instituto de Investigación en Ciencias Biomédicas, Centro Universitario de Ciencias de la Salud, Universidad de Guadalajara, Sierra Mojada 950, Guadalajara 44340, Jalisco, Mexico; 4Instituto de Nutrigenética y Nutrigenómica Traslacional, Centro Universitario de Ciencias de la Salud, Universidad de Guadalajara, Sierra Mojada 950, Guadalajara 44340, Jalisco, Mexico; nathaly.torrescas@academicos.udg.mx (N.T.-C.); juan.riverav@academico.udg.mx (J.J.R.-V.); roberto.rodriguez@academico.udg.mx (R.R.-E.)

**Keywords:** n-3 PUFA, inflammatory markers, obesity, fatty acids, DHA

## Abstract

The n-3 polyunsaturated fatty acids (PUFAs) can reduce inflammatory markers and may therefore be useful in obesity management. The aim of this study was to analyze the effect of supplementation with n-3 PUFAs on total fatty acid profile in red blood cells (RBCs), as well as biochemical and inflammatory markers, in subjects with obesity. The study consisted in a randomized placebo-controlled, double-blind clinical trial involving 41 subjects with obesity during a 4-month follow-up. Individuals were randomly assigned to two groups: n-3 PUFA supplementation (1.5 g fish oil) and placebo (1.5 g sunflower oil). Anthropometric, biochemical, dietetic, cytokine and total fatty acid profiles in RBCs were measured. Both groups increased their PUFA intake and DHA incorporation in RBCs. However, the placebo group showed a reduction in serum IL-8 and MCP-1 at the end of the study. A multiple linear regression model adjusted by body fat mass and sex showed that an increase in DHA in RBCs decreased the serum IL-8 levels in both study groups at the end of the study. Our results highlight the role of dietary DHA and n-3 supplementation usefulness in exerting beneficial anti-inflammatory effects.

## 1. Introduction

Obesity is a chronic disease characterized by the excessive accumulation of adipose tissue, which can be harmful for health [1,2]. The hypertrophy of adipocytes, particularly in visceral adipose tissue, has been linked to the development of metabolic diseases such as type 2 diabetes, nonalcoholic fatty liver, cardiovascular diseases, and certain types of cancer, among others [3,4]. According to the World Health Organization (WHO), 650 million adults have obesity, representing 13% of the global adult population [2].

Obesity management depends on the degree of overweight, associated chronic diseases, and individual factors such as age, sex, and genetic background. Currently, there are guidelines to assess patient health risks and treatment options, including pharmacology and bariatric surgery [4,5]; however, the first choice is a comprehensive lifestyle intervention that includes diet, physical activity, and behavioral therapy [5,6].

Regarding nutrition intervention, certain modifications such as calorie restriction and macronutrient distribution, including the ratio of omega 6 (n-6) and omega 3 (n-3) polyunsaturated fatty acids (PUFAs) have been suggested for obesity management [7,8]. It has been proposed that the optimal n-6:n-3 ratio should range between 1:1 to 5:1 to maintain a healthy balance. Nevertheless, Westernized diets are characterized by high consumption of n-6, which raises the n-6:n-3 ratio in a range of 10:1 to 20:1, which increases the risk of developing inflammatory diseases as obesity [9,10].

These essential fatty acids, linoleic acid (LA) and α-linolenic acid (ALA) must be consumed through the diet [11]. LA is the precursor for the synthesis of arachidonic acid (AA), a fatty acid in the n-6 series of PUFAs, while ALA is the precursor of eicosapentaenoic acid (EPA) and docosahexaenoic acid (DHA) in the n-3 series of PUFAs, although metabolites synthesized by the two types of PUFA are different and the enzymes involved in their metabolism are the same [12]. Increasing the n-3 PUFA relative content in the membrane phospholipids might be an important factor that counterbalances the presence of n-6 PUFA and thus the synthesis of eicosanoids derived from AA [13,14]. Thus, eicosanoids derived from the n-3 series of PUFA appear to have anti-inflammatory effects, while eicosanoids derived from the n-6 series of PUFA promote inflammation and seem to promote other pathologies as cardiovascular diseases; however, the evidence is limited by low numbers of events and risk of bias of the studies [15,16].

Both n-3 and n-6 PUFAs can regulate the transcription of genes involved in preadipocyte differentiation. Metabolites from the n-3 series of PUFA specifically act as ligands for peroxisome proliferator-activated receptors (PPARs) gamma (PPARγ) and delta (PPARδ), inducing fat cell differentiation and regulating lipoprotein lipase expression [10]. In this sense, supplementation with n-3 PUFA in subjects with obesity has been associated with a decrease in M1 macrophages in adipose tissue and proinflammatory markers [17].

Therefore, the aim of this study was to analyze the effect of supplementation with n-3 PUFA on total fatty acid profile in red blood cells (RBC), as well as biochemical and inflammatory markers, in subjects with obesity.

## 2. Materials and Methods

### 2.1. Subjects

A randomized placebo-controlled, double-blind clinical trial (ClinicalTrials.gov NCT04901052) was conducted from March 2018 to March 2019.

Subjects were randomly assigned to two groups using a table of random numbers: n-3 PUFA supplementation and placebo. Eighty-three participants were assessed for eligibility through flyers and social media invitations; however, only 58 met the inclusion criteria and the study was conducted on 41 participants (Figure 1).

The duration of the study was 4 months, and blood samples was taken at t = 0 (baseline) and t = 4 (4 months).

The inclusion criteria were as follows: subjects of both sexes, mestizos from west Mexico, (Mexican nationality), age from 30 to 50 years old, with obesity type I and II according to BMI (30–40 kg/m^2^), waist circumference (WC) ≥80 cm for women and ≥90 cm for men, Ref. [18] and sedentary lifestyle according to the WHO [19].

Exclusion criteria were pregnant or breastfeeding women, diagnosis of type 2 diabetes, cardiovascular disease, cancer, tobacco and alcohol consumption (≥40 g of alcohol per day for men and ≥20 g for women), and participants who consumed n-3 supplements, anti-inflammatory medications, or some type of lipid-lowering and anti-hyperglycemic drugs in the past year. All subjects signed an informed consent before enrollment. This study was approved by the Ethics and Biosafety Committee of the Health Sciences Center, of the University of Guadalajara, Jalisco, Mexico (registration CI-01219) and was in accordance with the WMA Declaration of Helsinki (1964) amended by the 64th WMA General Assembly Fortaleza, Brazil, October 2013 [20].

### 2.2. Nutritional Intervention

All study subjects were interviewed by a nutritionist to collect accurate hereditary family history, medical history, sociodemographic data, and dietary habits. A three-day dietary record was used to collect nutritional information (every month during the intervention). Food replicas from Nasco (United States^®^) were used to visualize portion size according to the Mexican Food Equivalents System, which allows greater accuracy when participants report the amount of food consumed of daily dietary intake. All dietary data were analyzed using the Nutritionist Pro™ software (Axxya Systems, Woodinville, WA, USA).

The nutritional intervention consisted of a hypocaloric diet with a 5:1 ratio of n6:n3 PUFA, and a 20% reduction in the total estimated energy intake was calculated using the Mifflin–St. Jeor equation based on the adjusted weight of participants every month at follow-up and considering a macronutrient distribution of 50% carbohydrates, 20% protein, and 30% lipids according to the estimated energy intake. In addition, a list of foods rich in n-3 PUFA was provided to emphasize their consumption during the study using the dietary traffic light labeling: green (daily consumption), yellow (frequent consumption 3–4 per week) and red (restricted consumption ≤ 2 per week). Reduced consumption of red meat, dairy products, and ultraprocessed foods (high in total sugar and saturated fat) was suggested, and the consumption of fish (at least two portions per week) and daily consumption of seeds rich in n-3 (chia and flaxseed), whole grains, fruits, and vegetables were emphasized (all the menu options that were given to the patients were adjusted by 5:1 n6:n3 PUFA ratio). The n-3 PUFA group was supplemented with 2 capsules per day of fish oil containing 1.5 g of n-3, of which 1000 mg was EPA and 500 mg DHA. The n-3 capsules were obtained from the same batch and a toxicity analysis was performed to verify their safety. The placebo group received 2 capsules per day containing 1.5 g of sunflower oil (0.9 g MUFA, 0.4 g PUFA, 0.1 g SFA). This oil has shown no effect on the n-3 profile [21]. The omega and placebo capsules were taken twice daily with breakfast to achieve the desired daily dose.

### 2.3. Adherence Evaluation

We analyzed adherence to the diet with a self-report provided by each participant, along with the three-day record analysis considering the adequation percentage of macronutrients and n6:n3 ratio, and finally, the percentage of adherence considering the self-reported value and dietary analysis (adequation percentage) from each subject, particularly considering compliance with n6:n3 ratio.

### 2.4. Anthropometric Measurements

The anthropometric measurements were performed after an 8–12 h overnight fast and with subjects wearing light clothing. Weight and body compartments were analyzed by electrical bioimpedance using an InBody 370 equipment (InBody 370, Biospace Co. Seoul, Republic of Korea). The waist circumference was measured with Lufkin Executive metal tape and the height with an SECA stadiometer with a precision of 0.5 cm (Rochester Clinical Research, New York, NY, USA). All measurements were standardized by three nutritionists who followed the same protocol during the study.

### 2.5. Biochemical Measurements

For biochemical analyses, peripheral vein blood was drawn at 7:00 to 9:00 am after an 8–12 h fasting period. The serum and plasma were separated by centrifugation at 3500 rpm for 15 min, and afterward, the liquid fraction was aliquoted, whereas a sample of RBCs (500 μL) was collected from the bottom of the EDTA tube and transferred to a microtube to which 10 μL of butylated hydroxytoluene (BHT) was added to protect the fatty acids from oxidation. Glucose levels and the lipid profile (total cholesterol, high-density-lipoprotein cholesterol (HDL-c), and triglycerides) were analyzed by dry chemistry using an automated Vitros 350 (Vitros 350 Analyzer, Ortho-Clinical Diagnostics, Johnson & Johnson Services, Inc., Rochester, NY, USA). Very low-density-lipoprotein cholesterol (VLDL-c) was estimated by dividing total triglycerides/5 and low-density-lipoprotein cholesterol (LDL-c) levels were calculated by the Friedewald formula as long as triglycerides did not exceed 400 mg/dL [22]. Insulin and adiponectin quantification was performed using an ELISA kit, catalogue number CT-600101A (International Diagnostics, Guadalajara, Jalisco, Mexico. S.A de C.V) following the supplier’s instructions.

### 2.6. Cytokine Levels

The quantification of proinflammatory and anti-inflammatory cytokines was performed using the Bio-Plex Pro™ Human cytokine Standard 17-Plex, Group I kit (catalogue number 10014905, Bio-Rad-Laboratories, Hercules, CA, USA). Following the supplier’s instructions, the plate was read immediately using the MAGPIX™ analyzer. IFNγ. IL-12, IL-13, IL-8, IL-6, MCP-1, MIP1β and adiponectin were the inflammatory markers that were measured.

### 2.7. Total Fatty Acid Profile of the RBCs

The fatty acid profile quantification of the RBCs was carried out by gas chromatography. For lipid extraction and derivation, a mixture of chloroform and methanol (2:1) containing 0.01% butylated hydroxytoluene (BHT) as an antioxidant was added to a 250 µL erythrocyte sample according to Folch’s technique [23].

All samples were analyzed in a gas chromatograph (GC) (Agilent Technologies, Santa Clara, CA, USA, 6850 network GS system) coupled with an injector (Agilent Technologies, 7083 Series) with a column for fatty acids (Durabond, DB-23 Agilent Technologies), a flame ionization detector with helium as gas carrier (0.7 cm^3^ min^−1^), and a temperature ramp (110 °C–220 °C).

The identification and quantification of the individual methyl esters were compared with known standards (Marinol^®^) and the complete fatty acid profile included: linoleic acid, arachidonic acid (AA), total n-6 PUFA, linolenic acid, EPA, docosapentaenoic acid (DPA), DHA, total n-3 PUFA, EPA/DHA ratio, and total PUFA. The percentage of fatty acids was determined by the normalized area. The results were expressed as percentage of each fatty acid in relation to the total fatty acids present in the sample.

### 2.8. Statistical Analysis

To calculate the sample size for this clinical trial, a formula that compares means between the treated and untreated group was considered with a power of 80% and an alpha value of 0.05 [24]. The estimated sample size was 19 subjects per group. The data obtained by Sugawara et al. according to changes in inflammatory cytokine concentrations in subjects who received a supplementation with n-3 PUFA were used as reference values [25].

A Shapiro–Wilk test was performed to evaluate the normality of the variables. Paired Student’s *t*-test or Wilcoxon test to evaluate changes between baseline and final evaluation periods was used depending on the normality of variables. Statistical differences between groups were analyzed using the independent Student’s *t*-test with delta values (final t = 4—baseline t = 0). Because multiple comparisons were made between groups, *p*-values were adjusted for a false-discovery rate using the Benjamini and Hochberg method in order to control for type I error, and 35% false discovery was expected [26]. A multiple linear regression model was used to analyze the contribution of DHA in RBCs, BFM and sex (independent variables) on the inflammatory markers (dependent variable: IFNγ. IL-12, IL-13, IL-8, IL-6, MCP-1, MIP1β and adiponectin), and all variables for the model were considered in t = 4 (end of study). The analyses were carried out with the SPSS v.20 software (IBM, Chicago, IL). To calculate the FDR, the R software v.4.2.2 was used [27]. A *p* value < 0.05 was considered statistically significant.

## 3. Results

### 3.1. Baseline Characteristics

Figure 1 shows the flowchart of the study. Eighty-three patients were screened for eligibility, 20 patients were not eligible (16 participants due to the presence of chronic disease, such as diabetes, cancer, hypertension, and hyper- or hypothyroidism and 4 patients who consumed omega-3 one month before the study), and 5 patients refused to participate and did not sign the informed consent. Fifty-eight eligible patients were randomly assigned to two study groups: n-3 PUFA group or placebo group and followed up for 4 months. Both received the same dietary intervention characterized by foods with enriched amounts of n-3 fatty acids (ALA, EPA, and DHA) at an n6:3 ratio was 5:1. Finally, 19 patients in the n-3 PUFA group and 22 patients in the placebo group completed the study (Figure 1). Notably, no statistically significant differences were found in any baseline anthropometric, biochemical, or nutritional data between the study groups (Table 1).

### 3.2. Changes in Nutritional Data at the End of the Study

At the end of the study, both groups showed a decrease in total energy intake and an increase in total dietary PUFA. Nevertheless, the placebo group showed an increase in dietary stearidonic acid and DHA. Both groups showed a decrease in total sugar intake; however, only the n-3 PUFA group was statistically significant (Table 2). Nonetheless, no statistically significant differences in any dietetic variable were found between the groups at the end of the study.

### 3.3. Changes in Anthropometric Data at the End of the Study

After 4 months of follow-up, the placebo group demonstrated significant improvements in anthropometric parameters of weight, BMI, WC, BFM, and fat. However, the n-3 PUFA group only showed an improvement in the WC parameter (Table 3).

After the 4 months of the study, differences between groups in weight, BMI, lean body mass and water were found in the placebo group; however, after the adjustment by Benjamini and Hochberg method, we did not find differences between groups (Table 3).

### 3.4. Changes in Biochemical Parameters at the End of the Study in Both Groups

Regarding the biochemical data of Table 4, no differences between baseline and final data were found among study groups. It was observed that the n-3 supplemented group had a difference in LDL-c levels in comparison with the placebo group (decrease 8.4 mg/dL vs. increase 11.0 mg/dL, respectively, *p* < 0.05); however, after adjustment by the Benjamini and Hochberg method, this significant difference was not found (Data shown in Appendix A).

### 3.5. Adherence to Diet during the Intervention

Participant adherence to the dietary intervention and nutritional recommendations were evaluated. In this regard, it was found that adherence decreased progressively towards the end of the study. During the 3rd and 4th months, adherence to the nutritional treatment decreased significantly (Table 5). In addition, a statistically significant difference was found between groups, showing a higher adherence percentage in the 3rd and 4th month in the placebo group compared to the n-3 PUFA group.

### 3.6. Changes in Inflammatory Markers at the End of the Study and Differences between Groups

Regarding inflammatory markers, a reduction in serum IL-8 and MCP-1 in the placebo group at the end of the intervention was found. Nevertheless, no significant changes were observed in the n-3 PUFA group. No significant differences were found between groups (Table 6). TNF-α and IL-6, were not detectable in any enrolled subjects. Although the cytokine analysis involved 17 analytes, only 7 are shown, because for the others, the sensitivity of the test marked them as undetectable (below the detection limit of the test used).

Regarding the incorporation of fatty acids in RBCs at the end of the study, we found an increase in ALA in the n-3 PUFA group (Figure 2A), an increase in DHA in both study groups (Figure 2B), and an increase in total n-3 PUFA only in the n-3 PUFA group (Figure 2C). The rest of the analyzed fatty acids did not show statistically significant differences at the end of the study (Table 7).

We found no significant differences regarding the incorporation of fatty acids in the RBCs at the end of the study between the groups.

When performing the regression model on the inflammatory markers, only IL-8 was significant. Finally, we constructed a model adjusted by BFM, age, and sex to show which variables influence serum IL-8 levels and we found that an increase in DHA in RBCs was associated with a decrease in serum IL-8 in both study groups (Table 8).

## 4. Discussion

This study showed that the diet intervention diminished energy intake and increased PUFA consumption in both study groups. Noteworthily, the remarkably higher consumption of PUFAs in both groups at the end of the study was independent of n-3 PUFA supplementation. Several studies have reported the benefits of diets with higher content of PUFAs, particularly higher intake of DHA, such as lower levels of cholesterol, triglycerides, and proinflammatory cytokines [28,29,30]. We did not find differences in the n-6:n-3 fatty acid ratio intake as reported by Guebre-Egziabher F et al.; however, they only analyzed LA and ALA, unlike our study, where more n-6 and n-3 fatty acids were considered for the calculation of the n-6:n-3 fatty acid ratio intake [28].

On the other hand, both groups showed improvements in anthropometric variables. Nevertheless, subjects in the placebo group reduced more body weight than individuals in the n-3 PUFA group. These differences could be attributed to the better adherence to the diet in the placebo group compared to the n-3 PUFA group. In fact, different research groups have found that a higher adherence to the diet is associated with a greater weight loss [31,32].

In this regard, adherence to the dietary intervention and nutritional recommendations by participants showed a progressive decrease towards the end of the study in both groups, although our adherence values were higher than those reported by Dansinger et al. [31] throughout their one-year intervention of four different diets. On the other hand, our placebo group showed a higher percentage of adherence at the 3rd and 4th month of follow-up compared to the n-3 PUFA group, which is in accordance with previous reports of higher adherence being associated with greater weight loss [31]. This difference in adherence can be attributed to the fact that a higher percentage of subjects from the n-3 group continued the intervention in the 3rd and 4th months during the month of December, which culturally has many festivities with high-calorie foods rich in sugar and fat, which could negatively impact the adherence to the meal plan in this particular group. In general, there is strong evidence of a dose–response relationship between adherence to a dietary plan and the expected clinical benefits [31]. Our findings might suggest that following a balanced dietary with a 5:1 ratio of n-6:n-3 and specific food recommendations as indicated at baseline might produce an equal or greater effect on lipid profile in RBCs and inflammatory markers when compared to a dietary plan with a lower degree of adherence but supplemented with n-3 PUFA. This is precisely the rationale behind the n-6:n-3 intake and the recommendations of their dietary sources of PUFA, rather than increasing the consumption of omega-3 alone [33].

The n-3 PUFAs have long been acknowledged for their anti-inflammatory and proresolution properties. This would be readily translated into lower serum inflammation markers in patients treated with fish oil [34]. However, the outcomes shown in our study highlight only two chemokines: IL-8 and MCP-1. Importantly, the circulating levels of these two markers have been elevated in subjects with a BMI >30 kg/m^2^ compared to those with a lower BMI (MCP-1, *p* < 0.02; IL-8, *p* < 0.01) [35]. In this regard, visceral fat is the component most often associated with metabolic alterations such as insulin resistance [36]. Nevertheless, in recent years, other research studies have reported that both subcutaneous and visceral fat might have an equal impact on metabolic health [37]. In fact, both IL-8 and MCP-1 have been strongly linked to insulin resistance for almost two decades [38], thus highlighting their potential role as therapeutic targets. Consequently, these markers can be influenced by any degree of weight loss, as shown in the placebo group by higher adherence to the dietary plan (65% versus 50%, Table 5), which might have played a significant role in this finding [39]. Notably, other cytokines, such as TNF-α and IL-6, known for their important role in insulin resistance, were not detectable in most enrolled subjects, so could not be included in the analysis [40]. Notwithstanding, no differences were found in the following parameters: glucose, plasma insulin, or HOMA-IR. Therefore, it is reasonable to think that if any modulation in the levels of these cytokines occurred, we consider that probably more time is necessary to observe additional metabolic health effects in the studied subjects.

Finally, our findings of fatty acid composition in RBCs are important evidence of the anti-inflammatory properties of n-3 PUFAs. On the one hand, the linear regression model we conducted illustrates the relevance of presenting a higher n-3 profile in RBCs, which has been considered a reliable marker. However, the importance of these outcomes not only lies in the anti-inflammatory effects of DHA through a ligand–receptor mechanism that can regulate the transcription of genes involved in the differentiation of preadipocytes and fat cells, including the regulation of lipoprotein lipase expression [10,41], but also in the fact that a higher percentage of this fatty acid has been found in other cell tissues, i.e., leukocytes, focused on supplementation with n-3 PUFAs [42]. This point is particularly interesting, since EPA and DHA in the phospholipid membrane of leukocytes serve as precursors for the synthesis of specialized proresolving mediators (SPMs), such as resolvins (E and D), maresins, and protectins, in the context of an established inflammatory process. The synthesis and circulating levels of these SPMs have been reported to occur in a dose-dependent manner, which leads to a reprogramming of peripheral blood cells; thus, SPMs have an important role in the regulation of persistent inflammation due to their capacity to inhibit microglia activation and reduce proinflammatory cytokines through the MAPK, NF-κB, PI3K/Akt, and caspase-3 signaling pathways [43]. However, the levels of these proresolution markers were not measured since they are beyond the scope of this research.

The main strength of the present study was the demonstration of the importance of including diet foods rich in n-3 PUFA and n-3 supplementation, which both may have positive effects on the incorporation of DHA in RBCs and might be associated with the decrease in inflammatory markers. This it also confirms that apart from the widely known effects of n-3 PUFA, a higher adherence to a 4-month dietary plan might provide stronger benefits for health improvement in obesity.

The limitations in our study were the small sample and the high dropout rate, which could be attributed to the low level of commitment to achieve a healthy lifestyle. It could also have been optimal to approach cytokine detection through ultrasensitive methodologies.

## 5. Conclusions

The beneficial effects of n-3 PUFAs depend primarily on the dietary n6:n3 ratio of polyunsaturated fatty acids along with an energy-restricted intervention. Based on the evidence of the role of PUFAs in the management of metabolic diseases, recommendations differ according to the study population, and for establishing an optimal dose, more research needs to be conducted. Furthermore, it would be beneficial to include groups with only a hypocaloric diet and another with only DHA supplementation and evaluating the relative contributions of these different factors, as it appears that diet adherence was the primary factor in successful weight loss.

However, an adequate consumption of DHA accompanied by weight loss due to calorie restriction is proposed as a therapeutic approach to reduce risks of complications in patients with obesity.

In this sense, it is crucial to highlight the importance of including dietary DHA, since it was observed that it is possible to increase the concentration of this fatty acid in the RBCs and is associated with having a positive impact by decreasing inflammatory markers such as MCP-1 and IL-8 in subjects with obesity in the context of higher adherence to a diet intervention.

## Figures and Tables

**Figure 1 healthcare-11-02333-f001:**
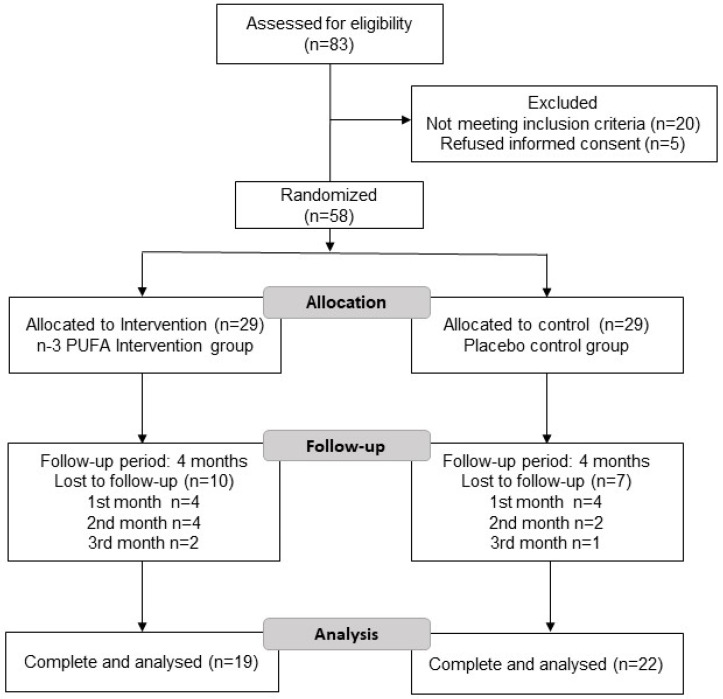
Flowchart of the subjects included in the study.

**Figure 2 healthcare-11-02333-f002:**
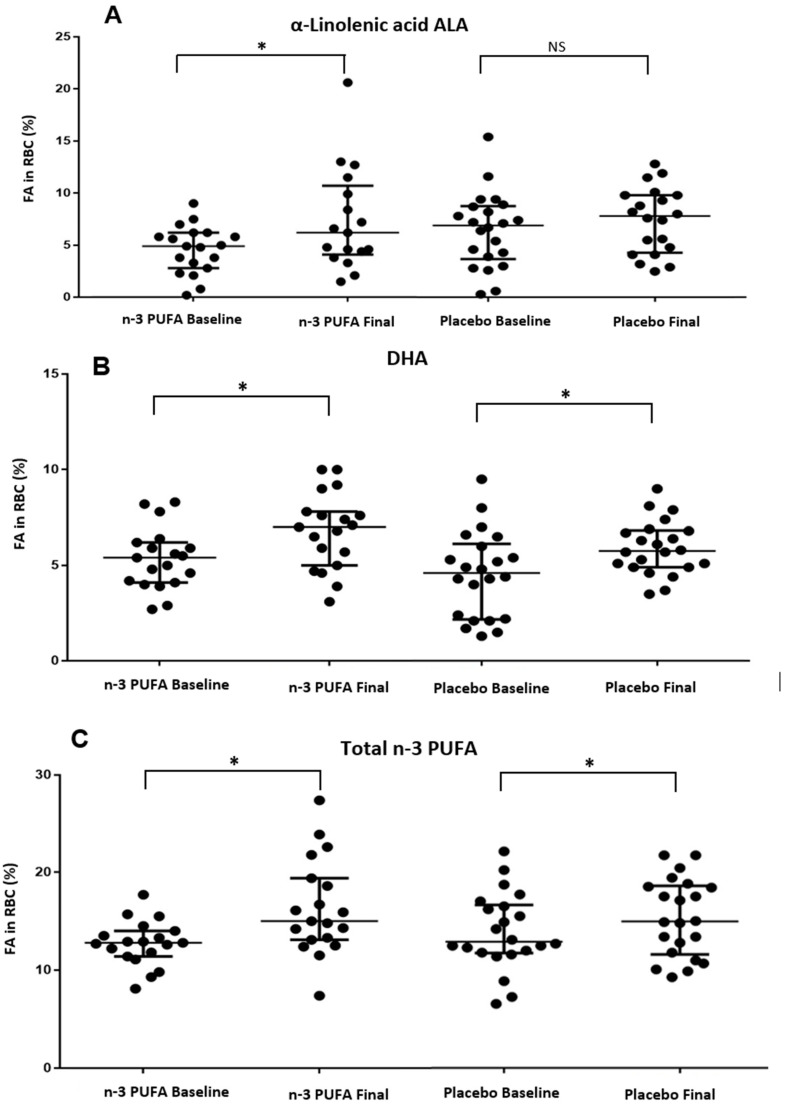
Changes of the fatty acids in RBC at the end of the study in both groups. * *p* value < 0.05. NS: non-significant difference. (**A**) Changes in α linolenic acid ALA in RBC during the intervention in study groups. (**B**) Changes in DHA in RBC during the intervention in study groups. (**C**) Changes in Total n-3 PUFA in RBC during the intervention in study groups.

**Table 1 healthcare-11-02333-t001:** Baseline (t = 0) clinical characteristics of study subjects.

Variable	n-3 PUFA Group Mean ± SD	Placebo Group Mean ± SD	*p* Value
Age (years)	39.5 ± 6.8	35.8 ± 6.6	0.092
Sex % (women/men)	55/50	45/50	
Nutritional			
Energy (kcal/day)	2555 ± 760.8	2148 ± 888.5	0.729
Protein (%)	17.0 ± 3.4	18.2 ± 2.7	0.312
Carbohydrates (%)	48.7 ± 11.3	42.3 ± 8.7	0.495
Fats (%)	35.8 ± 8.5	39.4 ± 7.9	0.276
SFAs (%)	12.2 ± 4.6	14.0 ± 3.8	0.155
MUFAs (%)	11.9 ± 3.4	14.1 ± 4.1	0.534
PUFAs (%)	3.4 ± 1.5	2.8 ± 1.4	0.440
Cholesterol (mg)	396.3 ± 230.2	442.4 ± 229.5	0.631
Linoleic (g)	21.2 ± 14.6	14.1 ± 10.0	0.413
20:4 AA	0.1 ± 0.1	0.1 ± 0.05	0.683
Linolenic (g)	1.7 ± 0.7	1.3 ± 0.6	0.759
EPA (g)	0.02 ± 0.04	0.02 ± 0.05	0.506
DHA (g)	0.06 ± 0.04	0.05 ± 0.04	0.395
22:5 DPA (g)	0.02 ± 0.01	0.01 ± 0.01	0.820
n6:n3 PUFA ratio	14:1	12:1	0.891
tFAs	0.7 ± 0.7	0.6 ± 0.7	0.867
Fiber (g)	22.2 ± 12.0	28.4 ± 16.4	0.179
Total sugar (g)	97.0 ± 38.2	80.5 ± 54.1	0.471
Anthropometric			
Weight (kg)	88.1 ± 9.8	92.4 ± 13.5	0.419
BMI (kg/m^2^)	32.8 ± 2.7	33.7 ± 32.3	0.372
WC (cm)	100.1 ± 9.6	102.5 ± 10.2	0.472
BFM (kg)	36.4 ± 5.5	38.8 ± 8.4	0.212
SM (kg)	28.4 ± 5.9	29.8 ± 7.2	0.954
Fat (%)	49.2 ± 9.9	48.5 ± 10.9	0.519
Lean body mass (kg)	41.7 ± 6.3	42.2 ± 7.5	0.901
Abdominal fat (%)	19.4 ± 2.1	19.5 ± 3.2	0.742
Water (kg)	37.0 ± 8.0	39.4 ± 8.5	0.699
Biochemical			
Glucose (mg/dL)	90.8 ± 8.3	102.3 ± 10.4	0.597
Insulin (µU/mL)	21.4 ± 14.9	24.1 ± 13.7	0.569
HOMA-IR	5.0 ± 4.1	5.8 ± 3.5	0.587
Total cholesterol (mg/dL)	181.6 ± 37.1	187.3 ± 35.5	0.711
HDL-c (mg/dL)	37.8 ± 6.3	39.3 ± 7.8	0.714
LDL-c (mg/dL)	126.1 ± 26.3	114.6 ± 23	0.767
VLDL-c (mg/dL)	28.2 ± 14.6	34.4 ± 1.9	0.294
TGs (mg/dL)	139.0 ± 68.5	169.1 ± 66.6	0.277
TC/HDL	5.0 ± 1.3	5.0 ± 1.4	0.881
LDL/HDL	3.2 ± 1.0	3.1 ± 0.9	0.743

These data are based on daily dietary intake. kcal/day: kilocalories per day, SFAs: saturated fatty acids, MUFAs: monounsaturated fatty acids, PUFAs: polyunsaturated fatty acids, mg: milligrams, g: grams, DPA: docosapentaenoic acid, tFAs: trans fatty acids, kg: kilograms, BMI: body mass index, WC: waist circumference, BFM: body fat mass, SM: skeletal muscle. HOMA-IR: homeostatic model assessment for insulin, HDL-c: high-density-lipoprotein cholesterol, LDL-c: low-density lipoprotein cholesterol, VLDL-c: very low-density lipoprotein cholesterol, TGs: triglycerides, IU/L: international units per liter, mg/dL: milligrams per deciliter. Comparisons between groups (intergroup) were analyzed with the independent Student’s *t*-test. Data expressed in mean ± standard deviation (SD).

**Table 2 healthcare-11-02333-t002:** Variation in nutritional variables during the study within each treatment group and between treatment groups.

Variable	n-3 PUFA Group				Placebo Group					
	Baseline t = 0	Final t = 4	*p*-Value ^a^	Δ	Baseline t = 0	Final t = 4	*p*-Value ^a^	Δ	*p*-Value ^b^	*q*-Value ^c^
Energy (kcal)	2555 ± 760	1590.7 ± 264	0.010	−964 ± 685	2148 ± 888	1597.5 ± 431	0.043	−550 ± 880	0.297	0.571
Protein (%)	17.0 ± 3.4	20.8 ± 2.3	0.077	3.7 ± 4.6	18.2 ± 2.7	18.7 ± 3.1	0.557	0.4 ± 2.8	0.065	0.406
Carbohydrates (%)	48.7 ± 11.3	46.0 ± 2.8	0.612	−2.6 ± 13.2	42.3 ± 8.7	40.8 ± 5.0	0.484	−1.5 ± 7.5	0.803	0.933
Lipids (%)	35.8 ± 8.5	35.5 ± 4.3	0.947	−0.3 ± 12.4	39.4 ± 7.9	43.4 ± 5.6	0.120	4.04 ± 8.7	0.368	0.634
SFAs (%)	12.2 ± 4.6	10.3 ± 1.6	0.345	−1.8 ± 4.8	14.0 ± 3.8	12.5 ± 2.0	0.230	−1.4 ± 4.1	0.840	0.933
MUFAs (%)	11.9 ± 3.4	12.8 ± 2.9	0.713	0.8 ± 5.9	14.1 ± 4.1	16.4 ± 4.2	0.192	2.2 ± 5.9	0.618	0.858
PUFAs (%)	3.4 ± 1.5	8.1 ± 3.0	0.015	4.6 ± 3.6	2.8 ± 1.4	10.1 ± 2.0	<0.001	7.2 ± 2.7	0.088	0.425
Cholesterol (mg)	396.3 ± 230.2	330.4 ± 115.0	0.532	−65 ± 262	442.4 ± 229.5	281.9 ± 101.1	0.058	−160 ± 275	0.466	0.706
Linoleic LA (g)	21.2 ± 14.6	12.1 ± 4.9	0.163	−9.1 ± 15.1	14.1 ± 10.0	15.7 ± 5.5	0.663	1.5 ± 12.7	0.111	0.426
20:4 AA (g)	0.1 ± 0.1	0.1 ± 0.06	0.856	−0.009 ± 0.1	0.1 ± 0.05	0.2 ± 0.1	0.085	−0.06 ± 0.1	0.091	0.425
Linolenic ALA (g)	1.7 ± 0.7	1.3 ± 0.7	0.352	−0.4 ± 1.1	1.3 ± 0.6	1.5 ± 0.8	319	0.2 ± 1.0	0.158	0.535
EPA (g)	0.02 ± 0.04	0.06 ± 0.06	0.186	0.04 ± 0.07	0.02 ± 0.05	0.1 ± 0.1	0.016	0.1 ± 0.1	0.282	0.571
DHA (g)	0.06 ± 0.04	0.1 ± 0.09	0.141	0.08 ± 0.1	0.05 ± 0.04	0.2 ± 0.1	0.003	0.2 ± 0.2	0.195	0.541
22:5 DPA (g)	0.02 ± 0.01	0.02 ± 0.01	0.185	0.005 ± 0.01	0.01 ± 0.01	0.02 ± 0.01	0.126	0.01 ± 0.02	0.596	0.851
n6:n3 PUFA ratio	14:1	9:1	0.480	−5:1	12:1	10:1	0.282	−2:1		
tFAs (g)	0.7 ± 0.7	0.3 ± 0.3	0.210	−0.3 ± 0.6	0.6 ± 0.7	0.1 ± 0.1	0.051	−0.4 ± 0.7	0.840	0.933
Total sugar (g)	97.0 ± 38.2	58.6 ± 17.1	0.047	−38.3 ± 40.7	80.5 ± 54.1	58.2 ± 21.4	0.100	−22.3 ± 45	0.444	0.693

SFA: saturated fatty acids, MUFAs: monounsaturated fatty acids, PUFAs: polyunsaturated fatty acids, AA: arachidonic acid, EPA: eicosapentaenoic acid, DHA: docosahexaenoic acid, DPA: docosapentaenoic acid, tFAs: trans fatty acids. Δ: final baseline data for all dietary variables. Data expressed in mean ± standard deviation (SD). t = 0 (time 0, baseline of the study), t = 4 (4 months, end of the study). ^a^ The intragroup comparisons (baseline vs. final) were performed with the paired Student’s *t*-test. ^b^ Delta comparisons between groups (intergroup) were analyzed with the independent Student’s *t*-test. ^c^ The *q*-values (intergroup) were adjusted with the Benjamini and Hochberg method to control for false-discovery rate. Data for fatty acid intake exclude intake from the capsules.

**Table 3 healthcare-11-02333-t003:** Variation in anthropometric parameters during the study within each treatment group and between treatment groups.

Variable	n-3 PUFA Group				Placebo Group					
	Baseline t = 0	Final t = 4	*p*-Value ^a^	Δ	Baseline t = 0	Final t = 4	*p*-Value ^a^	Δ	*p*-Value ^b^	*q*-Value ^c^
Weight (kg)	88.1 ± 9.8	87.5 ± 9.3	0.520	−0.6 ± 4.0	92.4 ± 13.5	88.5 ± 14.3	0.001	−3.9 ± 3.1	0.008	0.216 *
BMI (kg/m^2^)	32.8 ± 2.7	32.8 ± 3.0	0.866	−0.07 ± 1.9	33.7 ± 32.3	32.3 ± 3.8	0.001	−1.3 ± 1.0	0.013	0.216 *
WC (cm)	100.1 ± 9.6	95.5 ± 8.3	0.001	−4.5 ± 4.5	102.5 ± 10.2	96.5 ± 10.5	0.001	−6.0 ± 3.1	0.229	0.557
BFM (kg)	36.4 ± 5.5	36.6 ± 6.8	0.857	0.1 ± 3.6	38.8 ± 8.4	36.8 ± 9.5	0.010	−1.9 ± 3.1	0.062	0.406
SM (kg)	28.4 ± 5.9	27.9 ± 5.6	0.031	−0.5 ± 0.8	29.8 ± 7.2	28.6 ± 7.0	0.001	−1.1 ± 0.9	0.052	0.406
LBM (kg)	49.2 ± 9.9	48.9 ± 10.5	0.653	−0.5 ± 0.8	48.5 ± 10.9	50.4 ± 11.2	0.001	−1.8 ± 1.3	0.012	0.216 *
Fat (%)	41.7 ± 6.3	42.0 ± 7.4	0.546	0.3 ± 2.2	42.2 ± 7.5	41.8 ± 7.9	0.460	−0.3 ± 2.3	0.343	0.634
Abdominal fat (kg)	19.4 ± 2.1	19.4 ± 2.7	0.936	0.02 ± 1.3	19.5 ± 3.2	18.6 ± 3.7	0.023	−0.8 ± 1.5	0.977	0.981
Water (kg)	37.0 ± 8.0	36.9 ± 7.5	0.520	−0.07 ± 2.6	39.4 ± 8.5	37.8 ± 8.4	0.001	−1.5 ± 1.2	0.029	0.300 *

BMI: body mass index, WC: waist circumference, BFM: body fat mass, SM: skeletal muscle, LBM: Lean body mass, Δ: final baseline data for all anthropometric variables. Data expressed in mean ± standard deviation (SD). t = 0 (time 0, baseline of the study), t = 4 (4 months, end of the study). **^a^** The intragroup comparisons (baseline vs. final) were performed with the paired Student’s *t*-test. ^b^ Delta comparisons between groups (intergroup) were analyzed with the independent Student’s *t*-test. ^c^ The *q*-values (intergroup) were adjusted with the Benjamini and Hochberg method to control for false-discovery rate. * This value was a false discovery.

**Table 4 healthcare-11-02333-t004:** Variation in biochemical parameters during the study within each treatment group and between treatment groups.

Variable	n-3 PUFA Group				Placebo Group					
	Baseline t = 0	Final t = 4	*p*-Value ^a^	Δ	Baseline t = 0	Final t = 4	*p*-Value ^a^	Δ	*p*-Value ^b^	*q*-Value ^c^
Glucose (mg/dL)	90.8 ± 8.3	87.5 ± 4.7	0.092	−3.1 ± 7.5	102.3 ± 10.4	96.2 ± 10.7	0.104	−3.4 ± 9.3	0.911	0.969
Insulin (µU/mL)	21.4 ± 14.9	24.8 ± 22.9	0.289	3.3 ± 13.0	24.1 ± 13.7	22.2 ± 13.6	0.435	−1.8 ± 10.4	0.179	0.535
HOMA-IR	5.0 ± 4.1	5.7 ± 6.2	0.459	0.6 ± 3.3	5.8 ± 3.5	5.1 ± 3.4	0.279	−0.6 ± 2.6	0.206	0.542
TC (mg/dL)	181.6 ± 37.1	177.0 ± 32.1	0.442	−4.6 ± 24.8	187.3 ± 35.5	189.4 ± 24.1	0.701	2.1 ± 25.1	0.406	0.676
HDL-c (mg/dL)	37.8 ± 6.3	39.1 ± 9.9	0.496	1.2 ± 7.7	39.3 ± 7.8	39.1 ± 8.3	0.674	0.5 ± 5.6	0.728	0.910
LDL-c (mg/dL)	126.1 ± 26.3	117.2 ± 25.9	0.158	−8.4 ± 23.4	114.6 ± 23.7	125.6 ± 35.2	0.902	11.0 ± 28.4	0.030	0.300 *
VLDL-c (mg/dL)	28.2 ± 14.6	29.0 ± 13.7	0.471	1.7 ± 10.2	34.4 ± 1.9	31.0 ± 12.1	0.386	−3.1 ± 16.4	0.275	0.571
TGs (mg/dL)	139.0 ± 68.5	147.7 ± 68.4	0.478	8.7 ± 50.9	169.1 ± 93.2	143.1 ± 50.7	0.224	−25.9 ± 94.7	0.173	0.535

HOMA-IR: homeostatic assessment model of insulin sensitivity, TC: total cholesterol, HDL-c: high-density-lipoprotein cholesterol, LDL-c: low-density-lipoprotein cholesterol, VLDL-c: very low-density lipoprotein cholesterol, TGs: triglycerides. Δ: final baseline data for all biochemical variables. Data expressed in mean ± standard deviation (SD). t = 0 (time 0, baseline of the study), t = 4 (4 months, end of the study). **^a^** The intragroup comparisons (baseline vs. final) were performed with the paired Student’s *t*-test. ^b^ Delta comparisons between groups (intergroup) were analyzed with the independent Student’s *t*-test. ^c^ The *q*-values (intergroup) were adjusted with the Benjamini and Hochberg method to control for false-discovery rate. * This value was a false discovery.

**Table 5 healthcare-11-02333-t005:** Percentage of diet adherence in both study groups.

Variable	n-3 PUFA Group n = 19	Placebo Group n = 22	*p* Value
Adherence 1st month (%)	71.5 ± 17.8	78.9 ± 13.6	0.141
Adherence 2nd month (%)	66.5 ± 15.7	74.2 ± 17.8	0.135
Adherence 3rd month (%)	61.0 ± 19.4	73.8 ± 15.5	0.024
Adherence 4th month (%)	50.0 ± 19.7 *	65.5 ± 14.9 *	0.007

Data are presented as mean ± standard deviation of the adherence percentage and analyzed with Student’s *t*-test. * Statistically significant differences in adherence (1st month vs. 4th month) were defined as *p* < 0.05.

**Table 6 healthcare-11-02333-t006:** Variation in inflammatory markers during the study within each treatment group and between treatment groups.

Variable	n3 PUFA Group				Placebo Group					
	Baseline t = 0	Final t = 4	*p*-Value ^a^	Δ	Baseline t = 0	Final t = 4	*p*-Value ^a^	Δ	*p*-Value ^b^	*q*-Value ^c^
IFNγ (pg/mL)	10.1 ± 12.9	12.2 ± 17.0	0.860	2.0 ± 24.1	2.4 ± 2.3	1.1 ± 1.0	0.348	−1.3 ± 2.8	0.764	0.910
IL-12 (pg/mL)	0.6 ± 0.3	0.6 ± 0.3	0.420	−0.3 ± 1.5	0.7 ± 0.4	0.6 ± 0.5	0.423	−0.1 ± 0.1	0.286	0.571
IL-13 (pg/mL)	1.0 ± 1.3	0.7 ± 0.5	0.573	−0.3 ± 1.5	0.4 ± 0.3	1.7 ± 3.6	0.402	1.2 ± 1.3	0.268	0.571
IL-8 (pg/mL)	4.1 ± 1.5	3.7 ± 1.8	0.341	−0.3 ± 2.4	6.2 ± 2.8	4.2 ± 2.8	0.014	−2.0 ± 3.3	0.097	0.425
IL-6 (pg/mL)	0.6 ± 0.4	1.1 ± 0.9	0.513	0.5 ± 0.2	0.7 ± 0.4	0.1 ± 0.3	0.443	−0.6 ± 0.3	0.182	0.535
MCP-1(pg/mL)	71.1 ± 65.6	53.1 ± 43.3	0.197	−17.9 ± 48.5	54.7 ± 36.7	40.6 ± 25.9	0.033	−14.0 ± 27.4	0.760	0.910
MIP1β (pg/mL)	21.4 ± 13.1	17.2 ± 12.3	0.193	−4.2 ± 12.9	22.2 ± 21.2	16.2 ± 13.1	0.169	−5.9 ± 18.5	0.759	0.910
Adiponectin (μg/mL)	3.0 ± 2.9	2.2 ± 1.7	0.284	−0.8 ± 3.1	3.5 ± 4.2	3.1 ± 3.5	0.671	−0.4 ± 4.5	0.765	0.910

IFNγ: Interferon gamma, IL-12: Interleukin-12, IL-13: Interleukin 13, IL-8: Interleukin 8, IL-6: Interleukin 6, MCP-1: Monocyte chemoattractant protein 1, MIP1 β: Macrophage inflammatory protein-1 β, Δ: final baseline data for all inflammatory markers. Data expressed in mean ± standard deviation (SD). t = 0 (time 0, baseline of the study), t = 4 (4 months, end of the study). ^a^ The intragroup comparisons (baseline vs. final) were performed with the paired Student’s *t*-test. ^b^ Delta comparisons between groups (intergroup) were analyzed with the independent Student’s *t*-test. ^c^ The *q*-values (intergroup) were adjusted with the Benjamini and Hochberg method to control for false-discovery rate.

**Table 7 healthcare-11-02333-t007:** Variation in polyunsaturated fatty acid profile in RBCs during the study within each treatment group and between treatment groups.

Variable	n3 PUFA Group				Placebo Group					
	Baseline t = 0	Final t = 4	*p*-Value ^a^	Δ	Baseline t = 0	Final t = 4	*p*-Value ^a^	Δ	*p*-Value ^b^	*q*-Value ^c^
Linoleic acid (%)	7.8 ± 2.0	7.1 ± 1.6	0.209	−0.7 ± 2.3	8.8 ± 3.7	8.0 ± 2.2	0.386	−0.7 ± 4.0	0.981	0.981
AA (%)	8.3 ± 1.8	7.8 ± 2.4	0.487	−0.5 ± 3.2	7.6 ± 2.2	8.6 ± 1.8	0.085	0.9 ± 2.5	0.102	0.425
Total n-6 (%)	16.1 ± 3.5	14.9 ± 3.7	0.311	−1.2 ± 5.3	16.4 ± 4.4	16.7 ± 3.8	0.804	0.2 ± 5.0	0.361	0.634
α linolenic acid (%)	4.4 ± 2.3	7.3 ± 4.8	0.024	2.9 ± 4.8	6.1 ± 3.5	7.3 ± 3.1	0.248	1.2 ± 4.5	0.437	0.693
EPA (%)	1.5 ± 0.3	1.6 ± 0.3	0.420	0.08 ± 0.4	1.5 ± 0.3	1.5 ± 0.4	0.738	0.03 ± 0.5	0.732	0.910
DPA (%)	1.1 ± 0.5	1.4 ± 0.4	0.122	0.3 ± 0.8	1.1 ± 0.5	1.0 ± 0.2	0.323	−0.1 ± 0.5	0.234	0.557
DHA (%)	5.3 ± 1.6	6.7 ± 1.9	0.036	1.4 ± 2.7	4.5 ± 2.2	5.9 ± 1.4	0.035	1.4 ± 2.9	0.956	0.981
Total n-3 (%)	12.7 ± 2.8	16.3 ± 4.8	0.014	3.6 ± 5.8	15.2 ± 3.9	13.7 ± 3.9	0.321	1.4 ± 6.8	0.569	0.836
Total PUFAs (%)	28.8 ± 4.9	31.4 ± 6.6	0.269	2.5 ± 9.8	30.2 ± 6.2	32.0 ± 5.9	0.408	1.7 ± 9.6	0.874	0.950

AA: Arachidonic acid, n-6: omega 6, EPA: eicosapentaenoic acid, DPA: docosapentaenoic acid, DHA: docosahexaenoic acid n-3: omega 3, PUFAs: polyunsaturated fatty acids. Data expressed in mean ± standard deviation (SD). t = 0 (time 0, baseline of the study), t = 4 (4 months, end of the study). ^a^ The intragroup comparisons (baseline vs. final) were performed with the paired Student’s *t*-test. ^b^ Delta comparisons between groups (intergroup) were analyzed with the independent Student’s *t*-test. ^c^ The *q*-values (intergroup) were adjusted with the Benjamini and Hochberg method to control for false-discovery rate.

**Table 8 healthcare-11-02333-t008:** Variables associated with serum IL-8 levels at the end of the study (t = 4).

	B	CI	*p* Value
n-3 PUFA group (n = 19)			
DHA in RBCs t = 4 (%)	−0.426	−0.824–−0.028	0.038
BFM (kg) t = 4	−0.138	−0.275–0.000	0.051
Sex (1 woman, 0 men)	0.611	−1.473–2.695	0.535
Placebo group (n = 22)			
DHA in RBCs t = 4 (%)	−0.863	−1.682–−0.044	0.040
BFM (kg) t = 4	−0.028	−0.170–0.115	0.685
Sex (1 woman, 0 men)	2.253	−0.536–5.042	0.106

Linear regression model. Dependent variable: IL-8 (pg/mL) at the end of the study. BFM: body fat mass, DHA: docosahexaenoic acid, CI: Confidence interval. B: Regression coefficient (number of units that the dependent variable will increase or decrease for each unit that the independent variable increases or decreases).

## Data Availability

All data generated or analyzed during this study are included in this article. Further enquiries can be directed to the corresponding author.

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
