# Peer review of "A Balanced Dietary Ratio of n-6:n-3 Polyunsaturated Fatty Acids Exerts an Effect on Total Fatty Acid Profile in RBCs and Inflammatory Markers in Subjects with Obesity"

_healthcare, 2023, doi:10.3390/healthcare11162333_

Round 1

Reviewer 1 Report

The Author demonstrates in this artikle great effort and work in the field of dietary -energy restricted intervention with n-3 PUFA as  part of strategy in  preventive cardiology, with the scope to decrease inflamatory markers in sub. with obesity.The Abstract actually count 2011 words, instead of max. 200 w. Please if it is possible be little schorter max 200w. In abstract need to be present following part: 1) Background:  2) Methods: ; 3) Results:  and 4) Conclusion. The abstract should be a single paragraph and should follow the style of structured abstracts, but without headings. My suggestion to add paar words abot Ethical approvment for this clinical trial in Materoals and Methods. In Table 1. my suggestion is to add under lipid parametrs also two ateregonic indicators: TC/HDl, LDL/HDL.The tables, figures and generally part of Results , my opinion are excellent presented.Part of reference little be fresch up to date ultimate 5  year, with corrections for Journals in Italic font.

Some refrenece from Pubmed as usufull for add in part of refrences

The Effect of Omega-3 Fatty Acid Supplementation on Serum Adipocytokines, Lipid Profile and Biochemical Markers of Inflammation in Recreational Runners Aleksandra Żebrowska, Barbara Hall, Anna Stolecka-Warzecha, Arkadiusz Stanula, Ewa Sadowska-Krępa Nutrients. 2021 Feb; 13(2): 456. Published online 2021 Jan 29. doi: 10.3390/nu13020456   Select item 99203078. Omega-3 and Omega-6 Polyunsaturated Fatty Acid Intakes, Determinants and Dietary Sources in the Spanish Population: Findings from the ANIBES Study Marina Redruello-Requejo, María de Lourdes Samaniego-Vaesken, Ana M. Puga, Ana Montero-Bravo, Mar Ruperto, Paula Rodríguez-Alonso, Teresa Partearroyo, Gregorio Varela-Moreiras Nutrients. 2023 Feb; 15(3): 562. Published online 2023 Jan 21. doi: 10.3390/nu15030562 PMCID:  PMC9920307 Select item 65134559. Omega‐6 fats for the primary and secondary prevention of cardiovascular disease Lee Hooper, Lena Al‐Khudairy, Asmaa S Abdelhamid, Karen Rees, Julii S Brainard, Tracey J Brown, Sarah M Ajabnoor, Alex T O'Brien, Lauren E Winstanley, Daisy H Donaldson, Fujian Song, Katherine HO Deane Cochrane Database Syst Rev. 2018; 2018(7): CD011094 In the part of Conclusion the Authors gave pointed excellent the importante reasons for including dietyary dHA in diet. intervention in sense of reduction of inflam markers in prevention and treatment of obdesity subjects.

Author Response

Reviewer 1

The Author demonstrates in this article great effort and work in the field of dietary -energy restricted intervention with n-3 PUFA as part of strategy in preventive cardiology, with the scope to decrease inflammatory markers in sub. with obesity.

1.The Abstract actually counts 201 words, instead of max. 200 w. Please if it is possible be little shorter max 200w.

Answer: Thank you for your comment, the abstract was reduced to less than 200 words

2.In abstract need to be present following part: 1) Background:  2) Methods: 3) Results:  and 4) Conclusion. The abstract should be a single paragraph and should follow the style of structured abstracts, but without headings.

Answer. Thanks for your comment, the abstract was adapted according to your suggestions.

  1. My suggestion to add paar words abot Ethical approvement for this clinical trial in Materials and Methods. 

Answer: We include in the material and methods section that the Ethics and Biosafety Committee of the Health Sciences Center, which approved the clinical trial belongs to the University of Guadalajara, Jalisco, Mexico (Page 3, line 99)

  1. In Table 1. my suggestion is to add under lipid parameters also two ateregonic indicators: TC/HDL, LDL/HDL.

Answer: Thanks for your comment, we added in table 1 (page 6),  two aterogenic indicators the relation TC/HDL and LDL/HDL according to your suggestion.

5.The tables, figures and generally part of Results, my opinion are excellent presented.

Answer: Thank you.

  1. Part of reference little be fresch up to date ultimate 5 year, with corrections for Journals in Italic font.

Answer: Thanks to comment, we include updated references according to your pertinent suggestion and also we corrected the Italic font for the Journals. The  updated references have been highlighted.

7.Some reference from Pubmed as useful for add in part of references

The Effect of Omega-3 Fatty Acid Supplementation on Serum Adipocytokines, Lipid Profile and Biochemical Markers of Inflammation in Recreational Runners Aleksandra Żebrowska, Barbara Hall, Anna Stolecka-Warzecha, Arkadiusz Stanula, Ewa Sadowska-Krępa Nutrients. 2021 Feb; 13(2): 456. Published online 2021 Jan 29. doi: 10.3390/nu13020456   Select item 99203078.

Omega-3 and Omega-6 Polyunsaturated Fatty Acid Intakes, Determinants and Dietary Sources in the Spanish Population: Findings from the ANIBES Study Marina Redruello-Requejo, María de Lourdes Samaniego-Vaesken, Ana M. Puga, Ana Montero-Bravo, Mar Ruperto, Paula Rodríguez-Alonso, Teresa Partearroyo, Gregorio Varela-Moreiras Nutrients. 2023 Feb; 15(3): 562. Published online 2023 Jan 21. doi: 10.3390/nu15030562 PMCID:  PMC9920307 Select item 65134559.

Omega‐6 fats for the primary and secondary prevention of cardiovascular disease Lee Hooper, Lena Al‐Khudairy, Asmaa S Abdelhamid, Karen Rees, Julii S Brainard, Tracey J Brown, Sarah M Ajabnoor, Alex T O'Brien, Lauren E Winstanley, Daisy H Donaldson, Fujian Song, Katherine HO Deane Cochrane Database Syst Rev. 2018; 2018(7): CD011094 In the part of Conclusion the Authors gave pointed excellent the importante reasons for including dietyary dHA in diet. intervention in sense of reduction of inflam markers in prevention and treatment of obdesity subjects.

Answer: We include the references according to your suggestions in introduction and discussion sections (References 16, 28 and 31).

Reviewer 2 Report

Here are some suggestions that I hope may be useful to improve your manuscript.

General comments:

-          The wording in the introduction needs improvement. Consider shortening sentences for clarity and conciseness.

-          Please review the overall English language usage throughout the article.

-          Take a look at the table formats and ensure that there are no double lines in any of the tables.

Abstract:

-          The objective is lacking.

-          Considering that weight loss from the hypocaloric diet is not the main outcome, I suggest showing other relevant results in the abstract.

Introduction:

-          Line 38: I suggest revising to “…that could be harmful…”

-          Line 43-45: It is important to specify better the factors upon which the management of obesity depends, as well as the guidelines that are commonly used.

-          Line 55: Clarify that only LA and ALA are considered essential fatty acids.

-          The introduction should clearly outline the research gap and contribution of the study.

Methods:

-          The rationale for implementing a low-calorie diet and the potential confounding effects on isolating the impact of n-3 PUFA supplementation should be clarified. How was the effect of dietary changes distinguished from the effects of n-3 PUFA supplementation? Additionally, if the aim was to evaluate the n-6:n-3 ratio, why was a diet with a 5:1 ratio recommended? The study design seems inadequate to test the hypothesis.

-          Line 170-172: What was the calculated sample size?

Results and discussion:

-          Figure 2A: It would be helpful to mention if outliers were evaluated and how they were handled.

-          Lines 308 – 312: Did the participants follow the protocol during different months?

-          -360-362: This finding is widely accepted; please provide additional context or implications.

-          Strengthen the discussion on the strengths and limitations of the study.

Conclusions:

-          There is existing evidence for PUFA, EPA, and DHA recommendations.

-          374-376: To strengthen the conclusion, it would be beneficial to include groups with only a hypocaloric diet and another with only DHA supplementation. Additionally, evaluate the relative contributions of these different factors, as it appears that diet adherence was the primary factor for successful weight loss.

-          Line 379: Specify what inflammatory markers were analyzed (MCP1).

-          The inflammation markers evaluated are enough to reach this conclusion?

Author Response

Reviewer 2.

Here are some suggestions that I hope may be useful to improve your manuscript.

General comments:

1.The wording in the introduction needs improvement. Consider shortening sentences for clarity and conciseness.

Answer: Thanks for your comment, we revise the redaction of the introduction section to clarify sentences.

2.Please review the overall English language usage throughout the article.

Answer: Thanks for your comment, the English was revised.

3.Take a look at the table formats and ensure that there are no double lines in any of the tables.

Answer: Thanks for your comment, we confirm that there are no double lines in the tables.

Abstract:

1.The objective is lacking.

Answer: The objective was included in the abstract (Page 1, line 23-26)

2.Considering that weight loss from the hypocaloric diet is not the main outcome, I suggest showing other relevant results in the abstract.

Answer: We appreciate your comment, and we eliminated the results of weight loss and add relevant results according to inflammatory data (page 1, line 30-31).

Introduction:

1.Line 38: I suggest revising to “…that could be harmful…”

Answer: We correct this line according to your suggestion (line 39).

2.Line 43-45: It is important to specify better the factors upon which the management of obesity depends, as well as the guidelines that are commonly used.

Answer: The specify individual factors for obesity and other suggestions for obesity management like pharmacological and bariatric surgery. (Pages 1 and 2, lines 45-47).

3.Line 55: Clarify that only LA and ALA are considered essential fatty acids.

Answer: We clarify that in particular linoleic acid (LA) and α-linolenic acid (ALA) are essential fatty acids. Page 2, line 56.

4.The introduction should clearly outline the research gap and contribution of the study.

Answer. Thanks for your comment, we re-write some sentences to be more cleary of the contribution of the study in introduction section

Methods:

1.The rationale for implementing a low-calorie diet and the potential confounding effects on isolating the impact of n-3 PUFA supplementation should be clarified. How was the effect of dietary changes distinguished from the effects of n-3 PUFA supplementation? Additionally, if the aim was to evaluate the n-6:n-3 ratio, why was a diet with a 5:1 ratio recommended?

Answer: Thanks for your comment, first low-calorie diet was proposed because it is the mainly recommendation for subjects with obesity. For avoid possible confounders factors with n-3 supplementation capsules, we offer the same n-6:n-3 ratio, a 5:1 ratio (because as we mention in introduction section some authors propose 1:1 to 5:1 to  maintain a healthy balance) and the difference between the groups was the supplementation. We appreciate the feedback, it point will be considered in future studies

3.Line 170-172: What was the calculated sample size?

Answer: We add the sample size in the Methods section (Page 4, lines 182-183).

Results and discussion:

1.Figure 2A: It would be helpful to mention if outliers were evaluated and how they were handled.

Answer:  We consider the outlier in the final analysis because we didn’t have justification for not considering it, besides we have a modest sample size.

2.Lines 308 – 312: Did the participants follow the protocol during different months?

Answer: It was not possible to attend all the participants in the same month, however every patient full the 4 months intervention. The total intervention period was from March 2018 to March 2019 as mentioned in Methods section on page 2, line 83.

  1. 360-362: This finding is widely accepted; please provide additional context or implications.

Answer: We appreciate your comment. We included addiitional context about how the SPMs can inhibit microglia activation and reduce proinflammatory cytokines through the MAPK, NF-κB, PI3K/Akt, and caspase-3 signaling pathway. Page 13, lines 373-374.However we didn't measure because it was beyond the scope of this work, in future studies of resolution of inflammation should consider it.

4.Strengthen the discussion on the strengths and limitations of the study.

Answer: We have rewritten a part of the discussion strengthening the section on strengths and limitations (Page 13, lines 380-386).

Conclusions:

There is existing evidence for PUFA, EPA, and DHA recommendations.

  1. 374-376: To strengthen the conclusion, it would be beneficial to include groups with only a hypocaloric diet and another with only DHA supplementation. Additionally, evaluate the relative contributions of these different factors, as it appears that diet adherence was the primary factor for successful weight loss.

Answer: We appreciate your comment and we added your suggestion to the conclusion section (Page 13, lines 392-395)

2.Line 379: Specify what inflammatory markers were analyzed (MCP1).

Answer: We specify that the inflammatory markers were founded to have an impact in this study were MCP1 and IL-8 (Page 13, line, 402).

3.The inflammation markers evaluated are enough to reach this conclusion?

Answer: Thanks for your pertinent comment, we are aware that a broader panel of cytokines must be considered, the use of an ultrasensitive methodology for its detection was added in the limitations section; however, we highlight the cytokines found as keys in the inflammation process for the development of chronic diseases (Page 13, Lines 384-386).

Reviewer 3 Report

A balanced dietary ratio of n-6:n-3 polyunsaturated fatty acids 2 exerts an effect on erythrocyte membrane lipid profile and in-3 inflammatory markers in subjects with obesity

The file is in the format PDF/A and doesn’t allow suggestions to be made directly on the pdf file. I tried to comment as much as possible here.

General comments

The authors seek to clarify aspects relating to the importance of modifying people’s dietary habits (I think it is what is considered “intervention” in the report of the study) and the addition of n-3 FA supplements to the diet. As stated by the authors “The main strength of the present study was to demonstrate the importance of including in the diet foods rich in n-3 PUFA and n-3 supplementation, which both may have positive effects on the incorporation of DHA in RBCs and might be associated with the decrease of inflammatory markers.” And this is very important indeed, because both the quantity and quality of the diet are of paramount importance to combat obesity. The authors results seem to point out to importance of participants’ the adherence to the diet, thus stressing the importance of changing eating habits.

This paper has several problems that I detail below.

The authors state that they measured the effect of diet manipulation and n-3 fatty supplementation on the erythrocyte membrane lipid profile (title and caption of figure 2). However, the fatty acid analysis was made using total lipid extracts of RBC (line 154) and relates therefore to the total lipid composition of RBC. There isn’t any mention to lipid fractioning prior to methylation to separate phospholipids from TAGs. Please clarify this point and correct.

I would like to have some clarification on the place where the study was conducted because the authors mention “mestizos from West Mexico, age from 30 to 50 years old” (line 83), yet do not mention the nationality of the entity “ Ethics and Biosafety Committee of the Health Sciences Center”. The citation should read “according to the WMA Declaration of Helsinki (1964) amended by the 64th WMA General Assembly Fortaleza, Brazil, October 2013 (16)”

It is also not clear when the samples were taken, at the beginning of the study, after 1 month, 2 months and 3 months? This is also  not clear from the results where it seems that the measurements were only made at the beginning and end of the study (4 months?). This should be clarified in methods.

It also seem that the authors analysed the set of measured variables (both the nutritional value of the diet, and the biochemical variables measured in participants) by t-test without taking into account type I errors in null hypothesis testing when conducting multiple comparisons, and should correct their p-values with a suitable procedure, e.g. false discovery rate (FDR). It is also not clear what are “Changes in nutritional variables after the intervention in both study groups.” in table 1. I would prefer to have temporal information for the variables, perhaps t=0 for “baseline” and t= 4 months for “final”.

I don’t understand how the authors perform the “multiple linear regression (…) on the inflammatory markers”. There needs to a linear relationship between (a) the dependent variable and each of your independent variables (we are not told what they are), and (b) the dependent variable and the independent variables collectively. Did you measured “serum IL-8” more than twice (t=0 and t= 4 months)? If so, the analysis is inappropriate because measurements need to independent and here we would have a case of repeated measurements (on the same subjects). In table 8, what are the units of IL-8? and what is the B? If it is a variation in 2 points in time, you cannot apply a regression analysis. Please clarify this point. If necessary, apply other type of statistical analysis.

“Intervention” should be “study” or “test”, or does it have another meaning? Sometimes “intervention” also has the meaning of treatment. Please revise.

I consider that the report of the study has several inconsistencies that should be addressed before considering it for publication.

 Minor comments:

Abstract

The authors should correct the first sentence, revise the time and person of the verbs and insert commas so that sentences makes sense. The abstract should be re-written because it has several problems in its present form.

Line 20

The n-3 polyunsaturated fatty acids (PUFA) could can reduce inflammatory markers, and may  therefore, be useful in obesity management.

Line 21

This study analyze­s the effect of dietary supplementation with n-3 PUFA, considering an adequate n-6:n-3 PUFA ratio intake of 5:1, on red blood cells (RBC) lipid profile, biochemical, and inflammatory markers, in subjects with obesity.

The information must be better organised; the authors say in Line 21 that “supplementation with n-3 PUFA considering an adequate n-6:n-3 PUFA ratio intake (5:1)” and on Line 26 give a little more information on the same subject as “n-3 PUFA (1.5g n-3) and …”.

The “(1.5g n-3)” is also confusing, what do the authors mean by it? is it  the n-3 quantity of the n-3 supplementation in the “adequate n-6:n-3 PUFA ratio intake (5:1)”? Or the diet that was selected to have a n-6:n-3 PUFA ratio 5:1. This should be clarified.

Line 29

“At the end” means the end of the study?

Line 31

I find this line particularly puzzling. The authors write “Our results highlight the role  of dietary DHA since the proportion of n-6 and n-3 PUFA 5:1 as well as n-3 supplementation were useful to exert beneficial anti-inflammatory effects.” How many supplementation treatments were applied? In line 25, the authors state “(…) were randomly assigned into two groups: n-3 PUFA (1.5g n-3) and placebo (1.5g sunflower oil).”

Only on lines 114-116 do we learn that it was the diet that was adjusted to the “5:1 n6:n3 PUFA ratio” and n-3 were supplemented via capsules (treatment). Please revise the abstract and introduction.

Paragraphs

Sometimes the rationale behind the formation of paragraphs is not clear. For instance, line 47 shouldn’t have a paragraph break since the following paragraph deals with the same subject, namely “obesity management”. Please revise throughout the text.

The authors cite

11. Woodman, R.J.; Mori, T.A.; Burke, V.; Puddey, I.B.; Watts, G.F.; Beilin, L.J. Effects of Purified Eicosapentaenoic and Docosahexaenoic Acids on Glycemic Control, Blood Pressure, and Serum Lipids in Type 2 Diabetic Patients with Treated Hypertension. 422 Am J Clin Nutr 2002, 76, 1007–1015, doi:10.1093/ajcn/76.5.1007

when claiming, on line 59 that “Although metabolites synthesized by the two PUFAs families are different, the enzymes involved in their metabolism are the same [11].”

Citation 11 is incorrect because the subject of the cited article is not on fatty acid pathways. The authors should be also careful in how they formulate the sentences e.g. the metabolites are not “synthesized by the two PUFAs families” but from the two types of PUFA. A suitable reference should be supplied.

Could the authors also reference their sentence “In this sense, the increase of n-3 PUFA content in phospholipids could reduce the synthesis of eicosanoids from AA.” This is a rather bold statement since PUFA may also be recruited from other lipid classes (e.g. TAGs).

Line 71

I would remove the “Therefore” at the start of the sentence.

Line 73

I don’t understand what the authors mean by “(…) including biochemical and inflammatory markers in subjects with obesity.” Is it “the effect of diet supplemented with n-3 PUFA (…) ON the biochemical (what?) composition of RBC and inflammatory markers (…)”? Please clarify.

Line 81

what does it mean that “41 completed the intervention period”? Is it that the study was conducted on 41 participants. Are your results based on data from 41 participants? This should be clear, you should consider treatments as n-3 supplementation (n=19) and control (placebo) group (n=22).

Line 104

“besides” should be “and”. Check other places in the text where “besides” is used.

Figures and tables

I’m not sure if the flow-chart adds any important information to what is said in the methods’ section. Please supply captions that supply the necessary information to read the tables and figures. For instance, caption of table 2 should read as “Changes in the nutritional variables measured in the diet at the beginning (t=0) and after four months (t=4) of the study, of the two treatment groups. Data for fatty acid intake excludes treatment (FA supplements) intake. Data expressed in mean ± 1 (?) standard deviation (SD).”

The authors should correct the first sentence, revise the time and person of the verbs and insert commas so that sentences makes sense. The abstract should be re-written because it has several problems in its present form.

Author Response

Reviewer 3.

A balanced dietary ratio of n-6:n-3 polyunsaturated fatty acids 2 exerts an effect on erythrocyte membrane lipid profile and in-3 inflammatory markers in subjects with obesity

The file is in the format PDF/A and doesn’t allow suggestions to be made directly on the pdf file. I tried to comment as much as possible here.

 General comments

The authors seek to clarify aspects relating to the importance of modifying people’s dietary habits (I think it is what is considered “intervention” in the report of the study) and the addition of n-3 FA supplements to the diet. As stated by the authors “The main strength of the present study was to demonstrate the importance of including in the diet foods rich in n-3 PUFA and n-3 supplementation, which both may have positive effects on the incorporation of DHA in RBCs and might be associated with the decrease of inflammatory markers.” And this is very important indeed, because both the quantity and quality of the diet are of paramount importance to combat obesity. The authors results seem to point out the importance of participants' adherence to the diet, thus stressing the importance of changing eating habits.

This paper has several problems that I detail below.

1.The authors state that they measured the effect of diet manipulation and n-3 fatty supplementation on the erythrocyte membrane lipid profile (title and caption of figure 2). However, the fatty acid analysis was made using total lipid extracts of RBC (line 154) and relates therefore to the total lipid composition of RBC. There isn’t any mention to lipid fractioning prior to methylation to separate phospholipids from TAGs. Please clarify this point and correct.

Answer: We have added a lines in the methodology section to clarify the RBC isolation and preservation. Indeed, we assume that the total lipid analysis corresponds with the RBC membrane phospholipids, since the plasma was separated from the RBC fraction and the sample was taken from the bottom of the centrifuged tube. Thus, we did not proceed with a further chemical fractioning (Page 4, lines 145-150).

2.I would like to have some clarification on the place where the study was conducted because the authors mention “mestizos from West Mexico, age from 30 to 50 years old” (line 83), yet do not mention the nationality of the entity “Ethics and Biosafety Committee of the Health Sciences Center”. The citation should read “according to the WMA Declaration of Helsinki (1964) amended by the 64th WMA General Assembly Fortaleza, Brazil, October 2013 (16)”

Answer: We clarify that all participants were Mexican nationality (page 2, line 90), and that the ethics committee that evaluated them was from Mexico (page 3, line 99)

3.It is also not clear when the samples were taken, at the beginning of the study, after 1 month, 2 months and 3 months? This is also not clear from the results where it seems that the measurements were only made at the beginning and end of the study (4 months?). This should be clarified in methods.

Answer: We add in the methodology section that the duration of the study was 4 months, and blood samples was taken at t=0 (baseline) and t=4 (4 months) (page 2, lines 87-89).

4.It also seem that the authors analysed the set of measured variables (both the nutritional value of the diet, and the biochemical variables measured in participants) by t-test without taking into account type I errors in null hypothesis testing when conducting multiple comparisons, and should correct their p-values with a suitable procedure, e.g. false discovery rate (FDR). It is also not clear what are “Changes in nutritional variables after the intervention in both study groups.” in table 1. I would prefer to have temporal information for the variables, perhaps t=0 for “baseline” and t= 4 months for “final”.

Answer: Thank you for your suggestion. The p-values were adjusted with the Benjamini and Hochberg  method, to control for false discovery rate and the new adjusted  p-values were added in tables 2, 3, 4, 6 and 7.

5.I don’t understand how the authors perform the “multiple linear regression (…) on the inflammatory markers”. There needs to a linear relationship between (a) the dependent variable and each of your independent variables (we are not told what they are), and (b) the dependent variable and the independent variables collectively. Did you measured “serum IL-8” more than twice (t=0 and t= 4 months)? If so, the analysis is inappropriate because measurements need to independent and here we would have a case of repeated measurements (on the same subjects). In table 8, what are the units of IL-8? and what is the B? If it is a variation in 2 points in time, you cannot apply a regression analysis. Please clarify this point. If necessary, apply other type of statistical analysis.

Answer: Thank you for your observations. We have checked the assumption of linearity, the age and IL-8 were doing not have a linear relationship. Therefore, this variable was eliminated of the analysis and the new values are reported in table 8 in the revised manuscript. Besides, the variables included in lineal regression test were at t=4.

 6.“Intervention” should be “study” or “test”, or does it have another meaning? Sometimes “intervention” also has the meaning of treatment. Please revise.

Answer: Thanks for your comment we clarify this term in all the document.

7.I consider that the report of the study has several inconsistencies that should be addressed before considering it for publication.

Minor comments:

Abstract

1.The authors should correct the first sentence, revise the time and person of the verbs and insert commas so that sentences make sense. The abstract should be re-written because it has several problems in its present form.

Answer: We correct the first sentence and re-writte the abstract according to the reviewer's suggestion.

2.Line 20 The n-3 polyunsaturated fatty acids (PUFA) could can reduce inflammatory markers, and may therefore, be useful in obesity management.

Answer: Thanks for the comments, we improve this sentence in the abstract.

3.Line 21 This study analyze­s the effect of dietary supplementation with n-3 PUFA, considering an adequate n-6:n-3 PUFA ratio intake of 5:1, on red blood cells (RBC) lipid profile, biochemical, and inflammatory markers, in subjects with obesity.

Answer: Thanks for your comment, we included your suggestion.

4.The information must be better organised; the authors say in Line 21 that “supplementation with n-3 PUFA considering an adequate n-6:n-3 PUFA ratio intake (5:1)” and on Line 26 give a little more information on the same subject as “n-3 PUFA (1.5g n-3) and …”.

Answer: We clarify this point in the abstract, the diet was n-6:n-3 PUFA ratio intake (5:1) for both groups and the difference between them was the supplementation with 1.5g fish oil capsules.

5.The “(1.5g n-3)” is also confusing, what do the authors mean by it? is it  the n-3 quantity of the n-3 supplementation in the “adequate n-6:n-3 PUFA ratio intake (5:1)”? Or the diet that was selected to have a n-6:n-3 PUFA ratio 5:1. This should be clarified.

Answer: The diet was n-6:n-3 PUFA ratio intake (5:1) for both groups and the difference between them was the supplementation with 1.5g fish oil capsules.

6.Line 29 “At the end” means the end of the study?

Answer: We realized this change.

7.Line 31I find this line particularly puzzling. The authors write “Our results highlight the role of dietary DHA since the proportion of n-6 and n-3 PUFA 5:1 as well as n-3 supplementation were useful to exert beneficial anti-inflammatory effects.” How many supplementation treatments were applied? In line 25, the authors state “(…) were randomly assigned into two groups: n-3 PUFA (1.5g n-3) and placebo (1.5g sunflower oil).”

Answer: We improve this paragraph to better understanding of the idea.

8.Only on lines 114-116 do we learn that it was the diet that was adjusted to the “5:1 n6:n3 PUFA ratio” and n-3 were supplemented via capsules (treatment). Please revise the abstract and introduction.

Answer: Thanks for your comment, we clarify that the diet was adjusted to the “5:1 n6:n3 PUFA ratio” and n-3 were supplemented via capsules.

Paragraphs

9.Sometimes the rationale behind the formation of paragraphs is not clear. For instance, line 47 shouldn’t have a paragraph break since the following paragraph deals with the same subject, namely “obesity management”. Please revise throughout the text.

Answer: We integrate the ideas in one paragraph according to your suggestion.

10.The authors cite 11. Woodman, R.J.; Mori, T.A.; Burke, V.; Puddey, I.B.; Watts, G.F.; Beilin, L.J. Effects of Purified Eicosapentaenoic and Docosahexaenoic Acids on Glycemic Control, Blood Pressure, and Serum Lipids in Type 2 Diabetic Patients with Treated Hypertension. 422 Am J Clin Nutr 2002, 76, 1007–1015, doi:10.1093/ajcn/76.5.1007

when claiming, on line 59 that “Although metabolites synthesized by the two PUFAs families are different, the enzymes involved in their metabolism are the same [11].”

Citation 11 is incorrect because the subject of the cited article is not on fatty acid pathways. The authors should be also careful in how they formulate the sentences e.g. the metabolites are not “synthesized by the two PUFAs families” but from the two types of PUFA. A suitable reference should be supplied.

Answer: We correct the sentence and also we supply the reference for one in which the fatty acids pathway is described.

12.Could the authors also reference their sentence “In this sense, the increase of n-3 PUFA content in phospholipids could reduce the synthesis of eicosanoids from AA.” This is a rather bold statement since PUFA may also be recruited from other lipid classes (e.g. TAGs).

Answer: We have rewritten this sentence 60-62 to propose a more neutral statement (Page2, lines 60-62). Besides, two references have been added.

13.Line 71: I would remove the “Therefore” at the start of the sentence.

Answer: We eliminate the term “Therefore” as suggested by reviewer.

14.Line 73: I don’t understand what the authors mean by “(…) including biochemical and inflammatory markers in subjects with obesity.” Is it “the effect of diet supplemented with n-3 PUFA (…) ON the biochemical (what?) composition of RBC and inflammatory markers (…)”? Please clarify.

Answer:  We appreciate your comment, we improve the redaction of the objective in the abstract and introduction section to be more clear.

15.Line 81: what does it mean that “41 completed the intervention period”? Is it that the study was conducted on 41 participants. Are your results based on data from 41 participants? This should be clear, you should consider treatments as n-3 supplementation (n=19) and control (placebo) group (n=22).

Answer: We clarify that 41 subjects completed the study (line 87). The supplementation (n=19) and placebo group (n=22) is shown in Figure 1.

16.Line 104: “besides” should be “and”. Check other places in the text where “besides” is used.

Answer: This observation was attended as suggested by reviewer

17.Figures and tables: I’m not sure if the flow-chart adds any important information to what is said in the methods’ section. Please supply captions that supply the necessary information to read the tables and figures. For instance, caption of table 2 should read as “Changes in the nutritional variables measured in the diet at the beginning (t=0) and after four months (t=4) of the study, of the two treatment groups. Data for fatty acid intake excludes treatment (FA supplements) intake. Data expressed in mean ± 1 (?) standard deviation (SD).”

Answer: To clarify this point, we add t=0 (baseline data) and t=4 (4 months). And data for fatty acid intake excludes treatment (FA supplements) intake.

18.Comments on the Quality of English Language

The authors should correct the first sentence, revise the time and person of the verbs and insert commas so that sentences make sense. The abstract should be re-written because it has several problems in its present form.

Answer: This point was attended

Round 2

Reviewer 2 Report

I suggest improving the objective for clarity.

Author Response

I suggest improving the objective for clarified.

Answer: The objective was clarity as suggested by reviewer (abstract and introduction section).

The aim of this study was to analyze the effect of the supplementation with n-3 PUFA on total fatty acids profile in red blood cells (RBC), as well as biochemical, and inflammatory markers, in subjects with obesity.

Reviewer 3 Report

I thank the authors for their answers regard my previous concerns and the effort made to improve their manuscript. As it is, it still presents some important points that should be corrected and clarified. There are some aspects of the statistical analysis which are not correct and the authors seemed to simply have tried to show some work instead following the proper procedure.

One of my main concerns regards the fact that the fatty acid analyses is of RBC total lipids and not membrane lipids, as stated in the title and throughout the manuscript. In their reply, the authors state that “we assume that the total lipid analysis corresponds with the RBC membrane phospholipids, since the plasma was separated from the RBC fraction and the sample was taken from the bottom of the centrifuged tube.” And this is precisely my point, the authors analysed the fatty acids profiles of WHOLE cells (separated from plasma and serum as they state). As a minor comment on the supplied information, the authors should state the g force used to centrifuge the blood; the “g” force generated by a certain “rpm” depends on the rotor diameter, and cellules and cellular components are separated according to the g force applied to them. At 3500 rpm it is impossible to generate the necessary g (using any rotor in the scale of cm) to separate membrane cells from other celluar components, the authors simply collected whole cells. For example, to obtain RBC membranes, it is necessary to lyse the cells and collect the membrane by centrifugation at ca. 11000xg for some 30 minutes (at 4°C).

Thus, I must insist that the authors change the title and all references to “membrane fatty acids” to total fatty acids of the RBC, since no effort was made to separate, either by physical methods (e.g. selective centrifugation) or chemically, the membrane lipids from other lipid components of the cells.

The captions of tables and figures are largely incorrect and should be improved. The nutritional variables are given in kcal, g, etc, yet it is not said in what, daily dietary intake? Meal? This should be very clear. The whole manuscript should be carefully revised, the authors should check if they give all the necessary information to analyse each table or figure.

I don’t understand what the authors have done concerning the comparisons within groups and between groups and the calculation of p values taking false discovery rate into account. Considering Table 2 as an example, on the first version of the manuscript, the authors only gave values for variable variation (delta) within groups and the corresponding p-values. In the present version, the authors ascribe those values to “a” i.e. “The intragroup comparisons (baseline vs final)” and the p-values are followed by delta (inverted order relative to previous version). A new column is now added for p-values, which the authors state are “bComparisons between groups (intergroup)”. It is not said if this comparison is made at the end of at the beginning or the experiment, nor the variation (delta) is given. The preceding column delta is for “a”, I assume. The text of the manuscript (line 218), states that “Nonetheless, no statistically significant differences in any dietetic variable were found between the groups at the end of the study.” It seems the given comparisons were made for the values of variables at the end of the experiment. Please clarify.

I also don’t understand how false discovery rate (FDR) was taken into account. How much was your expectation of false discoveries, 20%, 25%? The authors state that the “p-values were adjusted with the Benjamini and Hochberg method, to control for false discovery rate” as previously reported in the methods section.

The authors reply to my comment (4) saying that the p-values were adjusted with the Benjamini and Hochberg  method, to control for false discovery rate and the new adjusted  p-values were added in tables 2, 3, 4, 6 and 7. Yet, the p-values in the tables are the same as in the previous version of the manuscript. Therefore, the given values are p-values and not adjusted since this method doesn’t give us adjusted values. I assume that the authors followed Benjamini-Hochberg procedure to control for FDR, in that case you may still give us the p-values but you must tell us which are still significant, since the variables must be ranked and significant differences are valid only for the variables which have a p-value smaller than the largest p-value that is less than its Benjamini-Hochberg critical value.

The authors should cite the authors whose methods they use:

Benjamini & Hochberg (1995) Controlling the False Discovery Rate: A Practical and Powerful Approach to Multiple Testing. https://rss.onlinelibrary.wiley.com/doi/10.1111/j.2517-6161.1995.tb02031.x

On Table 3, please check the caption that now states “Changes in anthropometric parameters at the end at end of the study between groups”. The caption should read “Variation of anthropometric parameters during the study within each treatment group and between treatment groups at the end of the study (if in fact that is what is happening here).

On line 238, the authors state “The asterisk (*) highlights the differences between groups.” This is information that should be given in the table and not in the body of text. Furthermore, I can’t find any asterisk on the table. The authors must carefully revise the text.

In the abstract the authors state (line 31) “a multiple linear regression model adjusted by body fat mass and sex showed that an increase of DHA in RBCs decreased the serum IL-8 levels in both study groups”. In methods, it is stated (line 190) “A multiple linear regression model was performed to analyze the contribution of variables on the inflammatory markers.” The reader needs to know which are the inflammatory markers, only IL-8 is reported, if so state that. The authors must state clearly in the methods, which are the dependent variables and which are the independent variables tested. If the authors are testing only IL-8 levels at the end of the study, the independent variables are values at the beginning of the study, the end, the variation? As replied to my comment number 5, age is not linearly related to IL-8 and was left out, this has to be stated in methods. Or you may refer to the variables in a certain table. Please give the “n” number of subjects in table 8.

 Previous comment number 2

The citation of the Declaration of Helsinki should be made correctly as I previously noted (line 100) should read “according to the WMA Declaration of Helsinki (1964) amended by the 64th WMA General Assembly Fortaleza, Brazil, October 2013 (16)”.

I thank the authors for their answers regard my previous concerns and the effort made to improve their manuscript. As it is, it still presents some important points that should be corrected and clarified.

One of my main concerns regards the fact that the fatty acid analyses is of RBC total lipids and not membrane lipids, as stated in the title and throughout the manuscript. In their reply, the authors state that “we assume that the total lipid analysis corresponds with the RBC membrane phospholipids, since the plasma was separated from the RBC fraction and the sample was taken from the bottom of the centrifuged tube.” And this is precisely my point, the authors analysed the fatty acids profiles of WHOLE cells (separated from plasma and serum as they state). As a minor comment on the supplied information, the authors should state the g force used to centrifuge the blood; the “g” force generated by a certain “rpm” depends on the rotor diameter, and cellules and cellular components are separated according to the g force applied to them. At 3500 rpm it is impossible to generate the necessary g (using any rotor in the scale of cm) to separate membrane cells from other celluar components, the authors simply collected whole cells. For example, to obtain RBC membranes, it is necessary to lyse the cells and collect the membrane by centrifugation at ca. 11000xg for some 30 minutes (at 4°C).

Thus, I must insist that the authors change the title and all references to “membrane fatty acids” to total fatty acids of the RBC, since no effort was made to separate, either by physical methods (e.g. selective centrifugation) or chemically, the membrane lipids from other lipid components of the cells.

The captions of tables and figures are largely incorrect and should be improved. The nutritional variables are given in kcal, g, etc, yet it is not said in what, daily dietary intake? Meal? This should be very clear. The whole manuscript should be carefully revised, the authors should check if they give all the necessary information to analyse each table or figure.

I don’t understand what the authors have done concerning the comparisons within groups and between groups and the calculation of p values taking false discovery rate into account. Considering Table 2 as an example, on the first version of the manuscript, the authors only gave values for variable variation (delta) within groups and the corresponding p-values. In the present version, the authors ascribe those values to “a” i.e. “The intragroup comparisons (baseline vs final)” and the p-values are followed by delta (inverted order relative to previous version). A new column is now added for p-values, which the authors state are “bComparisons between groups (intergroup)”. It is not said if this comparison is made at the end of at the beginning or the experiment, nor the variation (delta) is given. The preceding column delta is for “a”, I assume. The text of the manuscript (line 218), states that “Nonetheless, no statistically significant differences in any dietetic variable were found between the groups at the end of the study.” It seems the given comparisons were made for the values of variables at the end of the experiment. Please clarify.

I also don’t understand how false discovery rate (FDR) was taken into account. How much was your expectation of false discoveries, 20%, 25%? The authors state that the “p-values were adjusted with the Benjamini and Hochberg method, to control for false discovery rate” as previously reported in the methods section.

The authors reply to my comment (4) saying that the p-values were adjusted with the Benjamini and Hochberg  method, to control for false discovery rate and the new adjusted  p-values were added in tables 2, 3, 4, 6 and 7. Yet, the p-values in the tables are the same as in the previous version of the manuscript. Therefore, the given values are p-values and not adjusted since this method doesn’t give us adjusted values. I assume that the authors followed Benjamini-Hochberg procedure to control for FDR, in that case you may still give us the p-values but you must tell us which are still significant, since the variables must be ranked and significant differences are valid only for the variables which have a p-value smaller than the largest p-value that is less than its Benjamini-Hochberg critical value.

The authors should cite the authors whose methods they use:

Benjamini & Hochberg (1995) Controlling the False Discovery Rate: A Practical and Powerful Approach to Multiple Testing. https://rss.onlinelibrary.wiley.com/doi/10.1111/j.2517-6161.1995.tb02031.x

On Table 3, please check the caption that now states “Changes in anthropometric parameters at the end at end of the study between groups”. The caption should read “Variation of anthropometric parameters during the study within each treatment group and between treatment groups at the end of the study (if in fact that is what is happening here).

On line 238, the authors state “The asterisk (*) highlights the differences between groups.” This is information that should be given in the table and not in the body of text. Furthermore, I can’t find any asterisk on the table. The authors must carefully revise the text.

In the abstract the authors state (line 31) “a multiple linear regression model adjusted by body fat mass and sex showed that an increase of DHA in RBCs decreased the serum IL-8 levels in both study groups”. In methods, it is stated (line 190) “A multiple linear regression model was performed to analyze the contribution of variables on the inflammatory markers.” The reader needs to know which are the inflammatory markers, only IL-8 is reported, if so state that. The authors must state clearly in the methods, which are the dependent variables and which are the independent variables tested. If the authors are testing only IL-8 levels at the end of the study, the independent variables are values at the beginning of the study, the end, the variation? As replied to my comment number 5, age is not linearly related to IL-8 and was left out, this has to be stated in methods. Or you may refer to the variables in a certain table. Please give the “n” number of subjects in table 8.

Previous comment number 2

The citation of the Declaration of Helsinki should be made correctly as I previously noted (line 100) should read “according to the WMA Declaration of Helsinki (1964) amended by the 64th WMA General Assembly Fortaleza, Brazil, October 2013 (16)”.

Author Response

Reviewer 3

Comments and Suggestions for Authors

I thank the authors for their answers regard my previous concerns and the effort made to improve their manuscript. As it is, it still presents some important points that should be corrected and clarified. There are some aspects of the statistical analysis which are not correct and the authors seemed to simply have tried to show some work instead following the proper procedure.

One of my main concerns regards the fact that the fatty acid analyses is of RBC total lipids and not membrane lipids, as stated in the title and throughout the manuscript. In their reply, the authors state that “we assume that the total lipid analysis corresponds with the RBC membrane phospholipids, since the plasma was separated from the RBC fraction and the sample was taken from the bottom of the centrifuged tube.” And this is precisely my point, the authors analysed the fatty acids profiles of WHOLE cells (separated from plasma and serum as they state). As a minor comment on the supplied information, the authors should state the g force used to centrifuge the blood; the “g” force generated by a certain “rpm” depends on the rotor diameter, and cellules and cellular components are separated according to the g force applied to them. At 3500 rpm it is impossible to generate the necessary g (using any rotor in the scale of cm) to separate membrane cells from other cellular components, the authors simply collected whole cells. For example, to obtain RBC membranes, it is necessary to lyse the cells and collect the membrane by centrifugation at ca. 11000xg for some 30 minutes (at 4°C).

Answer: The reviewer is right, with the rpm at which we centrifuge the sample, it is not possible to separate membranes. For this reason, the term in the title and throughout the document was changed to the total fatty acids profile in RBC.

Thus, I must insist that the authors change the title and all references to “membrane fatty acids” to total fatty acids of the RBC, since no effort was made to separate, either by physical methods (e.g. selective centrifugation) or chemically, the membrane lipids from other lipid components of the cells.

Answer: The term “on membrane lipid profile in red blood cells (RBC)” was change to “total fatty acids profile of RBC” in all document.

The captions of tables and figures are largely incorrect and should be improved. The nutritional variables are given in kcal, g, etc, yet it is not said in what, daily dietary intake? Meal? This should be very clear. The whole manuscript should be carefully revised, the authors should check if they give all the necessary information to analyse each table or figure.

Answer: The captions of tables and figures were improved as suggested by the reviewer. Now, we mentioned that nutritional variables were according to daily dietary intake (line 107 and footnote table 1)

I don’t understand what the authors have done concerning the comparisons within groups and between groups and the calculation of p values taking false discovery rate into account. Considering Table 2 as an example, on the first version of the manuscript, the authors only gave values for variable variation (delta) within groups and the corresponding p-values. In the present version, the authors ascribe those values to “a” i.e. “The intragroup comparisons (baseline vs final)” and the p-values are followed by delta (inverted order relative to previous version). A new column is now added for p-values, which the authors state are “bComparisons between groups (intergroup)”. It is not said if this comparison is made at the end of at the beginning or the experiment, nor the variation (delta) is given. The preceding column delta is for “a”, I assume. The text of the manuscript (line 218), states that “Nonetheless, no statistically significant differences in any dietetic variable were found between the groups at the end of the study.” It seems the given comparisons were made for the values of variables at the end of the experiment. Please clarify.

Answer: Thank you for your comment, this point was clarified.

On the first version of the manuscript, we showed the p value, which was the intragroup comparisons (baseline vs final), and variable variation (delta) within groups only was indicated with (*) when the change was significant (p<0.05) but we did not put the p values. After your excellent remark, we add the p-value after adjustment by the Benjamini and Hochberg method. However, in this version of the manuscript to be clearer, we have included in the table 2, 3, 4, 6 and 7, aThe intragroup comparisons (baseline vs final), bThe intergroup comparisons (delta) and the corresponding p-values, and cThe q-values (intergroup comparisons) after adjustment by the Benjamini and Hochberg method.

I also don’t understand how false discovery rate (FDR) was taken into account. How much was your expectation of false discoveries, 20%, 25%? The authors state that the “p-values were adjusted with the Benjamini and Hochberg method, to control for false discovery rate” as previously reported in the methods section.

Answer: Thank for your observation. We expected a 35% of false discoveries and now was stated in the statistically section, in the line 192.

The authors reply to my comment (4) saying that the p-values were adjusted with the Benjamini and Hochberg method, to control for false discovery rate and the new adjusted  p-values were added in tables 2, 3, 4, 6 and 7. Yet, the p-values in the tables are the same as in the previous version of the manuscript. Therefore, the given values are p-values and not adjusted since this method doesn’t give us adjusted values.

Answer: We apologize for have been omitted the previous p-values. Now the p-values and the q-values were added in the tables. The q-values were calculated with R software using the function p.adjust (pvalues, method=“BH”).

I assume that the authors followed Benjamini-Hochberg procedure to control for FDR, in that case you may still give us the p-values but you must tell us which are still significant, since the variables must be ranked and significant differences are valid only for the variables which have a p-value smaller than the largest p-value that is less than its Benjamini-Hochberg critical value.

Answer: Thank you for your observation. We have added this information as supplementary material, and we have found five false discoveries (Tables 3 and 4). 

The authors should cite the authors whose methods they use:

Benjamini & Hochberg (1995) Controlling the False Discovery Rate: A Practical and Powerful Approach to Multiple Testing. https://rss.onlinelibrary.wiley.com/doi/10.1111/j.2517-6161.1995.tb02031.x

Answer: Thank you for your suggestion. The reference was added to the manuscript in the statistically procedures, in the line 192 (Reference 26).

On Table 3, please check the caption that now states “Changes in anthropometric parameters at the end of the study between groups”. The caption should read “Variation of anthropometric parameters during the study within each treatment group and between treatment groups at the end of the study (if in fact that is what is happening here).

Answer: This change was realized as suggested by the reviewer.

On line 238, the authors state “The asterisk (*) highlights the differences between groups.” This is information that should be given in the table and not in the body of text. Furthermore, I can’t find any asterisk on the table. The authors must carefully revise the text.

Answer: Thank you for your comment, this point was corrected.

In the abstract the authors state (line 31) “a multiple linear regression model adjusted by body fat mass and sex showed that an increase of DHA in RBCs decreased the serum IL-8 levels in both study groups”. In methods, it is stated (line 190) “A multiple linear regression model was performed to analyze the contribution of variables on the inflammatory markers.” The reader needs to know which are the inflammatory markers, only IL-8 is reported, if so state that. The authors must state clearly in the methods, which are the dependent variables and which are the independent variables tested. If the authors are testing only IL-8 levels at the end of the study, the independent variables are values at the beginning of the study, the end, the variation? As replied to my comment number 5, age is not linearly related to IL-8 and was left out, this has to be stated in methods. Or you may refer to the variables in a certain table. Please give the “n” number of subjects in table 8.

Answer: The dependent and independent variables were mentioned in statistical analysis (Lines 192 to 196), and the number of subjects “n” in table 8 was included. 

Previous comment number 2

The citation of the Declaration of Helsinki should be made correctly as I previously noted (line 100) should read “according to the WMA Declaration of Helsinki (1964) amended by the 64th WMA General Assembly Fortaleza, Brazil, October 2013 (16)”.

Answer: This suggestion was attended (Page 3, Line 99 to 100).
